# The Essentials on microRNA-Encoded Peptides from Plants to Animals

**DOI:** 10.3390/biom13020206

**Published:** 2023-01-19

**Authors:** Mélanie Ormancey, Patrice Thuleau, Jean-Philippe Combier, Serge Plaza

**Affiliations:** 1Laboratoire de Recherche en Sciences Végétales, CNRS/UPS/INPT, 31320 Auzeville-Tolosane, France; 2Epigenetics and Plant Development, Centre for Research in Agricultural Genomics (CRAG), UAB, Cerdanyola del Vallès, 08193 Barcelona, Spain

**Keywords:** microRNA, sORF, peptide, plant miPEP, animal miPEP

## Abstract

Primary transcripts of microRNAs (pri-miRNAs) were initially defined as long non-coding RNAs that host miRNAs further processed by the microRNA processor complex. A few years ago, however, it was discovered in plants that pri-miRNAs actually contain functional open reading frames (sORFs) that translate into small peptides called miPEPs, for microRNA-encoded peptides. Initially detected in *Arabidopsis thaliana* and *Medicago truncatula*, recent studies have revealed the presence of miPEPs in other pri-miRNAs as well as in other species ranging from various plant species to animals. This suggests that miPEP numbers remain largely underestimated and that they could be a common signature of pri-miRNAs. Here we present the most recent advances in miPEPs research and discuss how their discovery has broadened our vision of the regulation of gene expression by miRNAs, and how miPEPs could be interesting tools in sustainable agriculture or the treatment of certain human diseases.

## 1. Introduction

The adaptation of all living organisms to their environment requires the strict control of various biological processes that are essential for their growth, development, reproduction, and responses to stresses. This control is achieved through the modulation of signaling pathways by regulatory molecules that, *in fine*, activate or repress downstream target genes. In this context, microRNAs (miRNAs) play pivotal roles in growth, development, and stress responses. 

MiRNAs are small endogenous single-stranded RNAs (20 to 22 nucleotides) that are involved in post-transcriptional gene silencing in eukaryotes. They allow the downregulation of target genes by specifically triggering the degradation of their messenger RNAs (mRNAs) or by inhibiting their translation [1,2,3,4]. Most plant species have several hundred annotated miRNA genes. For example, the miRNA database miRbase (www.mirbase.org) contains 326 known miRNAs (accessed on 1 December 2022) in *Arabidopsis thaliana*, 604 in *Oryza sativa*, 247 in *Physcomitrella patens*, 594 in *Picea abies*, 2654 in *Homo sapiens*, 1978 in *Mus musculus*, 437 in *Caenorhabditis elegans*, and 469 in *Drosophila melanogaster*. These genes are grouped in families represented by a number in reference to similarities between their mature miRNA sequences. The various homologs within one multigene miRNA family are distinguished with different letters. Unlike animal miRNAs, which frequently target hundreds of genes, plant miRNAs usually have fewer than 10 targets, typically key regulators such as transcription factors (TFs), hormonal receptors, and nutrient sensors [5]. By downregulating key regulators, which in turn modify the expression of several genes, miRNAs thus act as developmental switches capable of modulating entire signaling networks. Since miRNA activity greatly influences physiological responses to developmental and environmental cues, it is obvious that any event that regulates miRNA activity could have drastic consequences on plant physiology and phenotypes.

For a long time, plant primary transcripts of miRNAs (pri-miRNAs) were annotated as long non-coding RNAs (lncRNAs). However, pioneering work on plants has highlighted the presence of small open frames (ORFs) in the 5’arm of these pri-miRNAs [6]. These ORFs can encode putative miRNA-encoded peptides (miPEPs) [6]. MiPEPs were further identified in animals [7]. Here, we highlight the most important published findings regarding miPEPs, from their discovery to their biological functions in both plants and animals, and we present the most recent data regarding the molecular mechanisms underlying miPEP activity. Finally, we discuss the potential application of miPEPs in agronomy and human therapeutics. 

## 2. MiPEP Discovery

Peptides are known to be involved in many processes including developmental regulation, acclimation to abiotic stress, and defense against pathogens [8,9,10,11,12] (Figure 1). The majority of known regulatory peptides in plants are derived from precursor proteins [13]. However, peptides that are directly translated from sORFs have also been reported [8,10]. Among them, those located in the 5’ region of pri-miRNAs, termed miPEPs, have recently received more attention [14,15,16]. Indeed, based on in-house and existing RACE-PCR-based annotations of pri-miRNAs of *M. truncatula* and *A. thaliana*, Lauressergues and colleagues (2015) performed an in silico analysis revealing the presence of at least one putative sORF in the 5’ region of *Mt*miR171b and *At*miR165a pri-miRNAs [6]. The functionality of these sORFs was validated for the first time in this study using *A. thaliana* and *M. truncatula* as model plants. Indeed, in both cases, the presence of endogenously expressed miPEPs was visualized by western blot and/or immunofluorescence using specific antibodies [6]. 

Since their discovery, the existence of miPEPs has been extended to various pri-miRNAs in several plant species as listed in Table 1 [17,18,19,20,21,22,23,24,25,26,27,28]. 

At the same time, the question of whether miPEPs exist in animals has arisen. The first description came from Razooky and co-workers (2017), who identified a miPEP called C17orf91 expressed from the pri-miRNA22 host gene [32]. MiPEP C17orf91 was upregulated upon viral infection but no associated function was reported. Later, several pri-miRNAs encoding miR34a, miR31, miR155, miR147b in mammals and miR8 and iab8 in *Drosophila* were described as capable of expressing miPEPs [31,33,34,35,36,37,38].

While it remains to be clarified in animals, several studies performed in plants on different miRNA genes have reported that the first ORF after the transcription start site is preferentially translated into a miPEP [6,19,23,39]. No common signature has been found among these different sORF-encoded peptides. However, so far, in plants, all tested miPEPs have been shown to act as an activator of their cognate miRNA expression contrasting with animals where only effects of sORF were detected [30,37,40].

## 3. MiPEP Functions 

### 3.1. In Plants

Several pieces of evidence suggest that miPEPs activate the expression of their miRNA genes. Indeed, the overexpression of *At*miPEP165a in a heterologous species (*Nicotiana benthamiana*), or the application of its synthetic version, increased the expression of both its corresponding pri-miRNA and the mature miRNA, and correlatively decreased the expression of miRNA target genes in *A. thaliana*. Similarly, the *M. truncatula* miPEP171b was able to increase its *Mt*pri-miR171b expression, suggesting that the function of miPEPs is conserved and not limited to a few species [6]. The positive effect of miPEPs on the accumulation of their respective pri-miRNAs was inhibited by cordycepin, a transcription inhibitor, suggesting that miPEPs induce this accumulation by increasing the transcription of their corresponding miRNA genes [6]. 

Due to the positive feedback that miPEPs exert on their corresponding pri-miRNAs in plants, miPEPs can be expected to exhibit diverse biological functions ranging from plant development to beneficial plant-microbial interactions or stress resistance, and could thus be considered as a natural alternative to pesticides and chemical fertilizers (Figure 1a). 

A study performed on grapevine was recently published in this context [23]. MiRNA171 family members are known to target genes involved in the formation and development of roots in different plants [6,41]. Chen and colleagues found that *VviMIR171* gene members were specifically expressed during the formation and development of grapevine (*Vitis vinifera*) adventitious roots [23]. When *Vvi*miR171d was overexpressed in *A. thaliana*, the plants displayed shorter primary roots, higher lateral root density, and earlier adventitious root development compared to wild-type (WT) plants. An in silico analysis predicted three putative sORFs in the 5’ region of *Vvi*pri-miRNA171d. Their respective transient overexpression in grape tissue culture plantlets showed that only the first pri-miRNA sORF enabled an increase in *Vvi*miR171d expression. In addition, when a construct containing the region from the *Vvi*miR171d promoter to the ATG start site of this sORF fused to the GUS gene was expressed in *N. benthamiana* leaves or grape tissue culture plantlets, GUS activity was observed. These data demonstrate that this sORF encodes a peptide, which was named *Vvi*miPEP171d1. Similar to what was previously described, when grape tissue culture plantlets were treated with synthetic *Vvi*miPEP171d1, *Vvi*miR171d expression specifically increased while the expression of miRNA target genes correlatively decreased. In addition, when grape plantlets were grown on a medium containing synthetic *Vvi*miPEP171d1, the number of adventitious roots significantly increased, indicating that the miPEP is able to regulate the formation and development of grapevine adventitious roots. This property appears specific to grapevines since *Vvi*miPEP171d1 had no effect on *A. thaliana* roots.

More recently, the same group characterized the function of two other miPEPs in grapevines, namely *Vvi*miPEP172b and *Vvi*miPEP3635b [25]. First, the authors identified *Vvi*miRNAs in grape tissue culture plantlets, whose expressions were modified during cold stress (4°C). They then selected *Vvi*miR172b and *Vvi*miR3635b for further analysis. Using an in silico approach, they identified six and four putative sORFs, respectively, in the 5’ region of the corresponding pre-miRNAs. They transiently expressed these sORFs in tissue culture plantlets independently and found that one ORF from each pre-miRNA was biologically active as it was able to increase the expression of its nascent pri-miRNA. They synthesized the corresponding miPEPs and, interestingly, their external application on grape tissue culture plantlets improved their tolerance to cold.

Another example illustrating the potentiality of miPEPs came from the study of the effect of *At*miPEP858a on *Arabidopsis* development [19]. *At*miR858 had previously been shown to downregulate the expression of different transcription factors such as *At*MYB11, *At*MYB12, and *At*MYB11, which regulate the phenylpropanoid pathway that sources the metabolites required for the biosynthesis of lignin and the production of many other important compounds such as flavonoids, coumarins, and lignans [42]. In addition, *At*miR858 modifies plant development by increasing root growth and accelerating flowering. By analyzing the region upstream of *At*pre-miR858a, the authors found three putative sORFs, of which one was shown to be translated in planta using reporter gene fusion assays and western blot experiments. This peptide, named *At*miPEP858a, increased the expression of both *At*pri-miR858a and mature *At*miR858 when exogenously applied to *Arabidopsis* seedlings; this also correlated with a downregulation of the expression of *AtMYB12* and its target genes, and phenotypically with an increase in root length. The effect of *At*miPEP858a was then confirmed via genetic approaches using both transgenic *Arabidopsis* plants overexpressing the miPEP and Cas9-edited *At*miPEP858a mutant plants. Thus, *At*miPEP858a-overexpressing plants exhibited longer main roots than WT plants, while edited mutant lines showed an inverted phenotype. Interestingly, the exogenous treatment of *At*miPEP858a-edited mutant plants with *At*miPEP858a complemented this phenotype. Compared to WT plants, *At*miPEP858a-overexpressing plants exhibited a reduction in anthocyanin accumulation as well as an increase in lignin content, together with enhanced expression of lignin biosynthesis genes. The reciprocal phenotype was observed in *At*miPEP858a-edited plants [19]. Very recently, the same group showed that a disulfated pentapeptide, named Phytosulfokine4 (PSK4), plays a key role in the growth and development of *At*miR858-dependent *Arabidopsis*, through auxin [28]. Interestingly, *At*miPEP858a positively regulates the expression of *PSK4* via *At*miR858a. The expression of *At*miR858a and *PSK4* is also positively regulated by the *At*MYB3 transcription factor through the direct binding of *At*MYB3 to its target promoters. *At*MYB3, whose expression is regulated by *At*miPEP858a/*At*miR858a, is a key component in *At*miPEP858a/*At*miR858a-PSK4-dependent plant growth and development [28]. Concomitantly to this study, the same authors showed that light directly regulates *At*miPEP858a accumulation in *Arabidopsis* and is necessary for *At*miPEP858a action. This light-dependent miPEP regulation requires the shoot-to-root mobile, light-mediated transcription factor, *At*HY5 [43]. Overall, the data place *At*miPEP858a at the crossroads of several biological processes, most likely through the regulation of its corresponding miRNA. 

MiPEPs can also modulate rhizospheric plant-microorganism interactions. For instance, an exogenous application of *Gm*miPEP172c specifically increases nodule numbers in soybean (*Glycine max*) when inoculated with *Bradyrhizobium diazoefficiens* and leads to an increase in *Gm*miR172c transcripts [21]. These results are in agreement with those previously observed by Wang et al., (2014), which show that *Gm*miR172c overexpression positively regulates nodulation in soybean through the repression of its target gene—the Apetala 2 (*Gm*AP2) transcription factor Nodule Number Control 1 (*Gm*NNC1)—which directly binds to the promoter of Early Nodulin 40 (*Gm*ENOD40) to repress its transcription [44]. Another example is the role played by *Mt*miPEP171b in arbuscular mycorrhizal symbiosis in *M. truncatula* [22]. Unlike other members of the *Mt*miPEP171 family, *Mt*miPEP171b stimulates arbuscular mycorrhizal symbiosis and positively regulates the expression of its corresponding *Mt*miR171b as well as the expression of *Mt*miR171b target *Mt*LOM1 (Lost Meristems 1). *Mt*miR171b is specifically expressed in root cells containing arbuscules and protects *Mt*LOM1 from being silenced by other *Mt*miR171 members through its mismatched cleavage site [22].

### 3.2. In Animals

With the miPEP description within *miR34a, miR31, miR155, and miR147b* genes in mammals and *miR8* and *iab8* genes in *Drosophila* (see above), it is now well established that pri-miRNAs can encode miPEPs in animal cells and, for some of them, their function and biology have even been documented. However, whether and how miPEPs regulate their corresponding pri-miRNA expression remains contradictory. To date, the only example in the animal literature describing a positive effect of a miPEP on the expression of its corresponding pri-miRNA is that of *Hs*miPEP133. *Hs*miPEP133 is a 133 amino acid peptide encoded by *Hs*pri-miR34a. *Hs*miPEP133 induces the expression of *Hs*pri-miR34a/miR34a which leads to the downregulation of *Hs*miR34a-targeted genes [33]. *Hs*miPEP133 is expressed in various healthy tissues but is downregulated in cancer cell lines and tumors. The overexpression of *Hs*miPEP133 indicates that the peptide acts as a human tumor suppressor *in cellulo* and in vivo by inducing apoptosis and inhibiting the migration and invasion of cancer cells. However, *Hs*MiPEP133 is mainly localized in mitochondria and not in nuclei as reported for plant miPEPs. It modulates a yet-to-be-defined signaling cascade that increases p53 transcriptional activity by disrupting mitochondrial function. Since *miR34a* is a direct target gene of the transcription factor p53, the latter upregulates both *Hs*miPEP133 and its corresponding *Hs*miR34a, most likely among a plethora of other p53 target genes. In addition, the authors showed that the positive feedback regulation of *Hs*miR34a by *Hs*miPEP133 can occur in both a p53-dependent and -independent manner, suggesting that miPEP133 can act through other molecular players [33]. More recently, Zhou and colleagues (2022) showed, in mice (*Mus musculus*), that *Mm*miPEP31 promotes the differentiation of regulatory T cells by repressing the expression of *Mm*miR31 in a sequence-dependent manner [38]. Interestingly, the authors showed that miPEP31 enters cells spontaneously and localizes to nuclei. The authors also demonstrate that miPEP31 negatively controls the expression of miR31, providing the first evidence that a miPEP can negatively control the expression of a miRNA gene. However, the mechanism involved seems different from that of miPEP133. Indeed, *Mm*miPEP31 binds to the *Mm*pri-miR31 promoter, induces the deacetylation of histone H3K27 (likely through the recruitment of a cofactor), and competes for the binding of an unknown transcription factor [38].

Although these two examples show that mammalian miPEPs are able to regulate their corresponding pri-miRNAs, either positively or negatively, animal miPEPs likely play other functions, which remain to be identified. The mechanism described in plants is probably not a general mechanism conserved in animal pri-miRNAs. Indeed, *Hs*miPEP200a, *Hs*miPEP155, *Hs*miPEP497, *Hs*MOCCI/MISTRAV, *Dm*miPEP8 and *Dm*MSAmiP do not reveal any effect on their corresponding pri-miRNA [30,31,34,35,36,37]. Furthermore, these miPEPs exhibit regulatory and biological functions uncoupled from their miRNA activity, acting either antagonistically to [31], in parallel with [37], or independently of [36], the miRNA pathway. 

To conclude this part, the studies described above show that while positive feedback regulation has been found in all plant miRNA genes studied so far, diverse miPEP effects have been reported in different animal model systems (Figure 1b), indicating that miPEP-dependent positive feedback regulation of miRNA genes is not a general mechanism that can be extended to all organisms.

## 4. What Features Underlie miPEP Activity?

### 4.1. MiPEP Entry into Cells

Given that an exogenous plant treatment with miPEPs induces significant phenotypic effects, one question remains unanswered: how do miPEPs enter plants? Recently, fluorescein-labeled *At*miPEP165a was shown to be internalized in *A. thaliana* roots by both endocytosis and passive diffusion [17]. Nevertheless, this miPEP did not enter the central cylinder and was not subsequently transported systemically. Its penetration was limited to the root peripheral zone, indicating that miPEPs might only act in a localized way. Accordingly, the application of miPEPs on leaves did not induce phenotypic changes to roots and, reciprocally, the application of miPEPs on roots did not induce changes to leaves. The entry of miPEPs through clathrin-mediated endocytosis was also recently reported by Badola and co-workers (2022) who used fluorescein-labeled *At*miPEP858a in *Arabidopsis* [28]. However, another study consisting of an exogenous application of a TAMRA-labeled miPEP, *Br*miPEP156a, to *Brassica rapa* seedlings indicates that this fluorescent peptide enters through the root system and accumulates predominantly in leaves [26]. This suggests that significant differences can be observed from one miPEP to another, perhaps due to their different physical/chemical properties.

In human, the cellular uptake of miPEPs has also been studied and it was shown that FITC-labeled miPEP155 efficiently entered into HEK293T cells and co-localized with endogenous miPEP155 (called P155) [31]. Another study in mouse showed that *Mm*miPEP31 behaves as a Cell Penetrating Peptide (CPP) both in vivo and in vitro [37]. FAM-labeled *Mm*miPEP31 enters across the cell membrane in an energy-independent manner thanks to its positively charged residues (5 Arg and 4 Lys on a 44-amino acid peptide), a common feature with other CPPs. This is however contrary to its nuclear transport, which appears to be energy-dependent. As observed in CPPs and transcription factors, a *Mm*miPEP31 structure prediction highlighted an α-helix structure [38]. 

Thus, these results show that miPEPs can behave as CPPs and constitute molecules acting in a non-cell-autonomous manner. The extent to which these peptides are endogenously secreted and able to function as long-range signaling molecules remains to be investigated.

### 4.2. MiRNA Genes Express Heterogeneous Populations of Transcripts in Plants

Pri-miRNAs are predicted to be localized to nuclei where the processing occurs. The processing is carried out by DCL1 in plants and Drosha/Patcha in animals. This however raises the question of how miPEP ORFs are translated.

In human, a study performed on the pri-miRNAs of genes encoding exonic miRNAs showed that some spliced pri-miRNA transcripts exhibit a cytoplasmic localization, consistent with a possible translation [45]. Moreover, as observed in plants, long non-coding miRNA host genes (pri-miRNAs), exhibit a complex gene structure, and are expressed as multiple transcript variants due to alternative promoter usage and/or alternative splicing [46]. This shows that miRNA genes produce many different transcripts, some of which lack the miRNA stem loop (Figure 2a).

Using Iso-Seq, RNA-Seq, and RACE-PCR data in *A. thaliana*, Lauressergues and co-workers (2022) recently showed that plant miRNA genes also express a heterogeneous population of transcripts in almost all studied cases: long canonical transcripts containing full-length sequences with the entire pre-miRNA (miRNA and miPEP sequences), and shorter transcripts, or, alternatively, spliced (AS) transcripts that only possess the miPEP sequence but not the entire pre-miRNA stem-loop sequence, i.e., the miRNA and miRNA* sequences (Figure 2b) [39]. Most short and AS transcripts appeared to be associated with the 60S ribosomal protein L18 (RPL18), suggesting that they are loaded into ribosomes. This is also the case for a few long pri-miRNA transcripts. These long transcripts are enriched within the nuclei, probably to generate mature miRNAs, compared to the cytoplasm where they are underrepresented. Short and AS transcripts, which are mainly associated with ribosomes and found in the cytoplasm, are most likely to generate miPEPs (Figure 2c). In vitro transcription/translation in wheat germ extracts reinforced the hypothesis that short transcripts are efficiently translated and constitute the main source of miPEPs [39].

### 4.3. Molecular Bases of miPEP Specificity in Plants

Although plant miPEPs appear to be poorly conserved across species, intriguingly, they fulfill an apparently very specific function by modifying the expression of their corresponding pri-miRNAs only, i.e., without disrupting the expression of other pri-miRNAs, even within the same miRNA family [6,39]. In this context, it is legitimate to wonder what molecular mechanisms underlie miPEP specificity, and more generally their functions. Recent data have highlighted that the sORF-encoding miPEP (miORF) itself plays a pivotal role in miPEP responses and specificity [39]. Thus, its deletion leads to an absence of pri-miRNA induction by the corresponding miPEP. Moreover, swapping the sequence of *M. truncatula* miORF171b with that of *A. thaliana* miORF319a in *Mt*pri-miRNA171b prevents *Mt*miPEP171b activity, whereas *At*miPEP319a becomes active to positively modulate the expression of *Mt*pri-miR171b. Similarly, the insertion of an artificial miPEP sequence (not present in the plant genome) in *Mt*pri-miR171b makes this pri-miRNA activatable by the corresponding artificial miPEP. Furthermore, miORF localization appears important for miPEP-induced activation. Indeed, when the ORF is placed on the 3’ arm of the microRNA within the pri-miRNA, no miPEP-induced activation could be observed. However, duplicating the ORF in multiple copies on the 5’ arm increased the miPEP-induced response. These observations suggest an interplay between the miPEP and its corresponding ORF that is important for the miPEP response. Consistently, FRET–FLIM (Förster resonance energy transfer–fluorescence lifetime imaging microscopy) and ITC (isothermal titration chemistry) approaches focusing on *Mt*miPEP171b indicate that the miPEP is indeed at proximity or probably interacts with its pri-miRNA via the miORF [39].

Taken together, these data shed light on how non-conserved miPEPs can perform specific regulatory functions in their host species only. 

### 4.4. MiPEP Conservation

Unlike miRNAs, which are highly conserved among plant species, miPEP sequences appear to be much more variable and do not possess any common signature [22,39]. With the exception of *At*miPEP156a, *At*miPEP164a, and *At*miPEP165a which exhibit some conservation, miPEP sequences are generally not conserved within *Brassicaceae* [6,39,47]. An exogenous application of *At*miPEP156a and *At*miPEP167a, highly (≈ 90% identity) and poorly (≈50–70% identity) conserved miPEPs among *Brassicaceae*, respectively, revealed that only *At*miPEP156a positively upregulated its pri-miRNA in all plants tested—i.e., *A. thaliana*, *Brassica rapa* and *Brassica oleracea* [39]. In the same way, Chen and collaborators (2020) have shown that an exogenous application or an overexpression of *Vvi*miPEP171d1 from grapevines does not affect root development in *A. thaliana* [23]. Similarly, an application of *At*miPEP171c, the *A. thaliana* miPEP ortholog of *Vvi*miPEP171d1, does not induce any phenotype change in grapevines, whilst it does promote the growth of *A. thaliana* lateral and adventitious roots. These miPEPs are therefore only active on their plant of origin due to the poor conservation of their sequences among plant species. Importantly, both conserved and non-conserved miPEPs retain the same potential to upregulate the transcription of their respective microRNA gene. This illustrates that a lack of sequence conservation does not signify that they are non-functional. Therefore, non-conserved miPEPs are species-specific regulators.

Only a few miPEPs have been reported in animals so far, but the same observation was made with regard to conservation. Indeed, while miPEP133 is only conserved in primates, miPEP155, miPEP497, and miPEP31 are conserved between primates and mice [30,38], confirming that the sORF-encoded peptides are less conserved than proteins [48]. Similarly, human miPEP200 or *Drosophila* miPEP8 are not conserved, whereas the *D. melanogaster* micropeptide MSAmiP, encoded by the previously thought non-coding RNA called *male-specific abdominal* (*msa*), shows homology within several distant *Drosophila* species [36]. 

### 4.5. First Insight into the Mechanisms of miPEP Activity in Animals 

Few mechanistic advances have been made in animals. Kang and co-workers (2020) performed a miPEP133 interactome in human and found only a few proteins interacting with it [33]. In particular, they revealed its interaction with the mitochondrial chaperone HSP9A resulting in the disruption of the interaction of HSP9A with other proteins, explaining the antitumor activity of miPEP133. Also in human, miPEP155 has been described to interact with the chaperone heat shock cognate protein 70 (HSC70) [31]. Similar to miPEP133, miPEP155 disrupts the HSC70-HSP90 machinery which affects major histocompatibility complex class II-mediated antigen presentation and T-cell priming [31]. In mouse, no protein partner has been described for miPEP31. However, electrophoretic mobility shift assays (EMSA) revealed that this peptide can bind to a specific DNA motif and disrupt the binding of proteins present in nuclear extracts, revealing a regulation induced by competitive interaction [38]. 

Although limited, these data nonetheless give an idea of the multiplicity of functions potentially supported by miPEPs in animals.

## 5. Perspectives

While several miPEPs have been described in plants and their mode of action is beginning to be deciphered, questions remain to be better clarified: how is their specificity achieved, and do all plant miPEPs use the same molecular machinery. Lauressergues et al., (2015) suggested that they control miRNA genes at the transcriptional level [6]. Therefore, it is probable that they are in contact with the transcriptional machinery and either regulate RNA polymerase II activity and/or that of the mediator complex (Figure 3). The level of activation is relatively modest (about twice), suggesting that miPEPs slightly (and/or transiently?) increase either transcription initiation and/or elongation. Another important question is how and why many different miPEPs (differing in length and sequence) retain the ability to interact with these protein complexes. Some possible answers can be found in the literature. First, a large number of peptides, some of which are structured and folded and some of which are not, are known to interact with proteins [49]. Secondly, there are examples of peptide/protein interactions in certain organisms that differ from those commonly described. In bacteria for example, Oligopeptide-binding Protein A (OppA) from *Lactococcus lactis* binds peptides ranging from 4 to 35 amino acids in length with little sequence selectivity to deliver many different peptides to the oligopeptide permease (Opp) [50]. Sequence specificity between peptides and target proteins, which is usually ensured by peptide side chains, is absent in OppA/peptide interactions. In contrast, hydrogen bonds form between the protein and the peptide backbone itself, covering a stretch of 5 residues—which explains why OppA retains the ability to interact with many different peptide sequences [50]. Another example is illustrated by the Ca^2+^ pump SERCA which interacts with different regulatory peptides [51]. Importantly, although no strong conservation has been detected within their primary amino acid sequences, regulatory peptides retain a similar secondary structure [52].

The mediator complex has been extensively studied in plants. It is a central integrator of transcription conserved among species. Composed of a large number of subunits, it has the ability to bind to many proteins or complexes, such as transcription factors and RNA polymerase II, to integrate regulatory signals at the transcriptional level [53]. It is also capable of making structural changes that, in turn, increase its ability to bind to many other proteins. In fact, it is thought that the mediator complex can interact with all transcription factors (hundreds to thousands) present in genomes. Interestingly, in plants, it has been described to integrate several abiotic stresses [53], revealing its pleiotropic functions, also observed for miPEPs. Therefore, like for transcription factors, miPEPs interaction with different subunits/regions of the mediator complex is a possible hypothesis to explain why so many different miPEPs act similarly at the transcriptional level. 

Finally, although there is evidence that miPEPs control pri-miRNA expression at the transcriptional level [6,19,23], this does not preclude their involvement in pri-miRNA processing. Indeed, it has been shown that pri-miRNA transcription and processing are actually coupled [54]. Consequently, it is possible that miPEPs influence pri-miRNA processing in parallel. Therefore, the identification of miPEP partners may be key to understanding how miPEPs auto-regulate the expression of their miRNAs and could shed light on the underlying mechanisms.

One important limitation of plant miPEP discovery remains the lack of precise annotations of pri-miRNAs in plants. This problem has been improved only very recently for most *A. thaliana* pri-miRNAs via sequencing technologies [39]. However, miPEP identification remains largely challenging for some plant species, including crops or weeds, due to the lack of genomic and transcriptomic data. To overcome this problem, a set of miPEPs from *A. thaliana* have been tested for their ability to modulate *Arabidopsis* root growth [18]. This work focused on those that had the strongest effects, miPEP397a and miPEP164b which positively and negatively affected total root development, respectively. Homologs of these peptides have been sought in *B. oleracea* (*Bo*miPEP397a; cabbage) and the weed *Barbarea vulgaris* (*Bv*miPEP164b). Interestingly, *Bo*miPEP397a was able to increase root length in cabbage as well as foliar surface by positively regulating the expression of its corresponding pri-miRNA, whereas the opposite effect was observed for *Bv*miPEP164b on *B. vulgaris,* again through upregulation of its corresponding pri-miRNA [18]. 

In the animal field, it remains to be established how many microRNA genes are capable of producing miPEPs. To date, all described miPEPs originate from intergenic miRNA genes. Unlike plants, the vast majority of animal miRNAs are intronic and embedded in coding genes. It remains to be established whether and how many microRNA genes are able—or unable—to produce miPEPs. Finally, all known examples suggest that animal miPEPs use diverse molecular mechanisms to act, which requires a thorough characterization of the molecular functions involved for each miPEP. 

## 6. Conclusions

One of the major challenges for the coming years is to reduce the use of herbicides and fertilizers through the use of natural molecules that are safer for people and more respectful of the environment. In addition, more plants become resistant due to the inappropriate and intensive use of herbicides. MiPEP technology could be an alternative for dealing with these ecological problems. Since miPEPs are highly specific, one can imagine using a cocktail of several peptides to improve crop growth, yield as well as resistance to biotic stresses in the context of the environment becoming more and more unpredictable due to global environmental changes. By targeting miRNAs with specific miPEPs, one could also imagine reducing weed growth. 

However, many questions remain unsolved. For instance, how do miPEPs communicate with the transcriptional machinery, and, more precisely, through which mechanisms is miPEP-induced pri-miRNA expression regulated? Is the only interaction between miPEPs and their corresponding miORFs sufficient? Are regulatory proteins necessary for miPEP action and/or specificity, and, if so, which ones? 

In the animal field, it appears that the regulation of pri-miRNA expression by miPEPs is not a conserved mechanism and varies from one pri-miRNA to another. Given that many diseases in humans are due to aberrant expression of miRNA genes, it is tempting to anticipate that the use of miPEPs might represent novel therapeutic treatments. However, this will require a perfect understanding of the mechanisms involved.

## Figures and Tables

**Figure 1 biomolecules-13-00206-f001:**
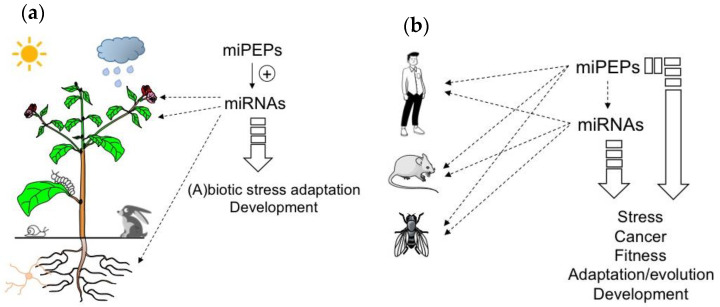
MicroRNA-encoded peptides (miPEPs) regulate many biological functions both in plants and animals. (**a**) The ability of plant miPEPs to positively regulate the expression of their respective pri-miRNAs is described for several miPEPs and plant species. (**b**) Conversely, in animals, the regulation of pri-miRNAs by miPEPs is less clear. MiPEPs frequently act independently.

**Figure 2 biomolecules-13-00206-f002:**
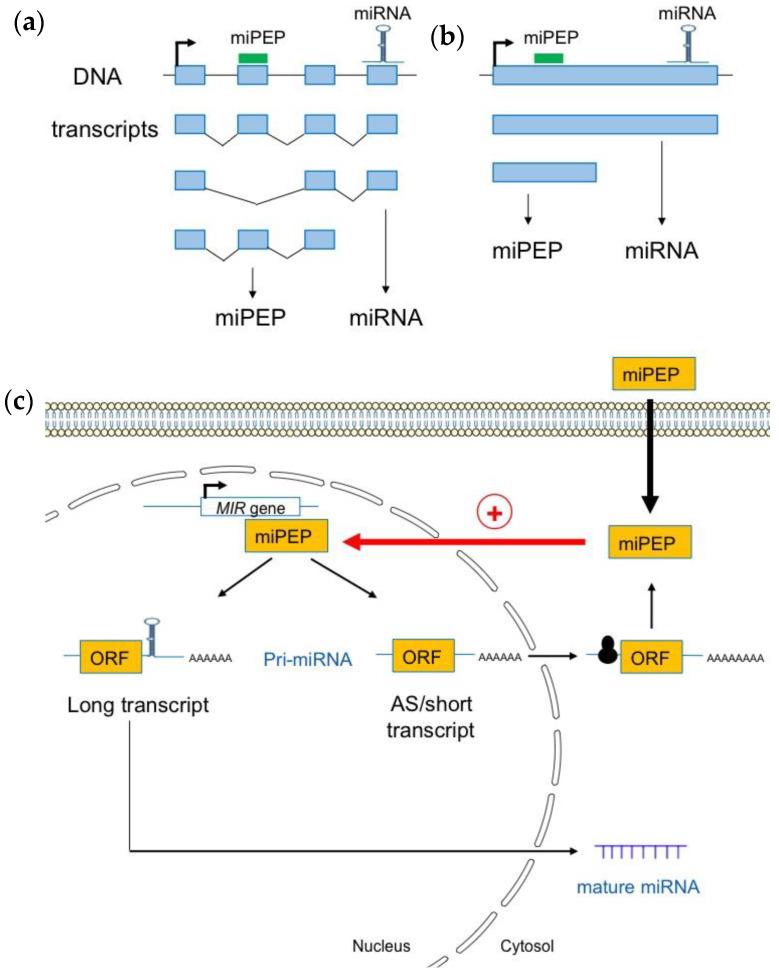
Plant pri-miRNAs are processed into a heterogeneous population of transcripts. (**a**) Pri-miRNAs are produced from alternative splicing (**b**) or alternative transcriptional termination sites. (**c**) Most short and alternatively spliced (AS) pri-miRNA transcripts are localized in the cytoplasm where they interact with the 60S ribosomal protein L18 (RPL18), suggesting that they are loaded into ribosomes for translation. Pri-miRNA transcripts containing the miRNA stem loop are enriched within nuclei where they can be used as templates to generate mature miRNAs (adapted from [39]).

**Figure 3 biomolecules-13-00206-f003:**
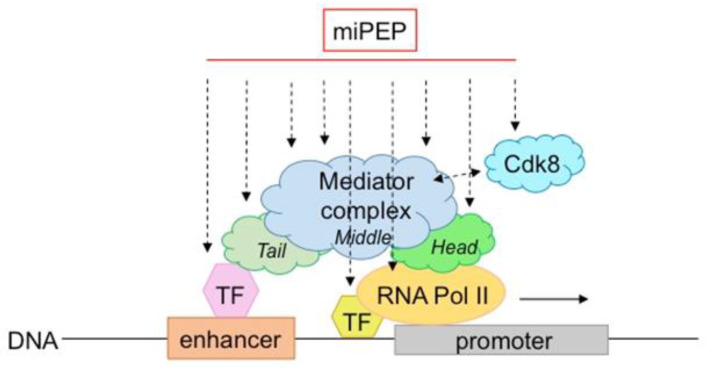
Model of possible miPEP interactions/crosstalk within the transcriptional machinery. MiPEPs might interact with different subunits/regions of the mediator or RNApolII complexes.

**Table 1 biomolecules-13-00206-t001:** List of miPEPs (and their embedded miRs) described in the literature (and miRbase) both in plants and animals.

Organism	MiPEP (miR)	MiPEP Size	In Vivo miPEP Detection	Effect on the Corresponding Pri-miRNA	Regulation of miRNA Targets	Regulated Biological Functions	Ref
**Plants**							
*Arabidospsis thaliana*	AtmiPEP165a (ath-miR165a)	18	GUS reporter gene expression and wb	Upregulation	Downregulation of *HD-ZIP III PHAVOLUTA, PHABOLUSA, REVOLUTA*	Stimulation of main root growth; Acceleration of the inflorescence stem appearance and of the flowering time; Inhibitory effect on total root growth	[6,17,18]
*Arabidopsis thaliana*	AtmiPEP858a (ath-amiR858a)	44	GUS reporter gene expression and wb	Upregulation	Downregulation of *MYB* transcription factor *AtMYB12*	Flavonoid biosynthesis and plant development	[19]
*Arabidopsis thaliana*	AtmiPEP164b (ath-miR164b)	29	N/A	Upregulation	Downregulation of *NAC1*, *NAC4*, *NAC5*, *CUC1* and *CUC2*	Inhibitory effect on total root growth	[18]
*Arabidopsis thaliana*	AtmiPEP397a (ath-miR397a)	7	N/A	Upregulation	Downregulation of *LAC2*, *LAC4* and *LAC17*	Stimulation of total root growth	[18]
*Dimocarus Longan* Lour	N/A	50	N/A	Upregulation	Downregulation of *HD-ZIP IIIATHB15*	Embryogenesis	[20]
*Glycine max*	GmmiPEP172c (gma-miR172c)	16	N/A	Upregulation	Downregulation of *AP2* transcription factor *NODULE NUMBER CONTROL 1*	Increase in nodule number	[21]
*Lotus japonicus*	LjmiPEP171b (lja-miR171b)	22	N/A	Upregulation	N/A	Increase in mycorrhization rate	[22]
*Medicago truncatula*	MtmiPEP171b (mtr-miR171b)	20	GUS reporter gene expression and wb	Upregulation	Upregulation of *GRAS* transcription factor *LOST MERISTEMS 1* (*LOM1*)	Reduction of lateral root development and increase in mycorrhization rate	[6,22]
*Medicago truncatula*	MtmiPEP171a (mtr-miR171a)	10	N/A	N/A	Downregulation of *LOM1*	Decrease in mycorrhization rate	[22]
*Medicago truncatula*	MtmiPEP171c (mtr-miR171c)	7	N/A	N/A	Downregulation of *LOM1*	Decrease in mycorrhization rate	[22]
*Medicago truncatula*	MtmiPEP171d (mtr-miR171d)	6	N/A	N/A	Downregulation of *LOM1*	Decrease in mycorrhization rate	[22]
*Medicago truncatula*	MtmiPEP171e (mtr-miR171e)	23	N/A	N/A	Downregulation of *LOM1*	Decrease in mycorrhization rate	[22]
*Medicago truncatula*	MtmiPEP171f (mtr-miR171f)	5	N/A	N/A	Downregulation of *LOM1*	Decrease in mycorrhization rate	[22]
*Oryza sativa*	OsmiPEP171i (osa-miR171i)	31	N/A	Upregulation	N/A	Increase in mycorrhization rate	[22]
*Solanum lycopersicum*	SlmiPEP171e (slymiR171e)	19	N/A	Upregulation	N/A	Increase in mycorrhization rate	[22]
*Vitiis vinifera*	VvimiPEP171d1 (vvi-MIR171d1 *)	7	GUS reporter gene expression	Upregulation	Downregulation of *scarecrow-like VvSCL27*	Adventitious root formation	[23]
*Vitis vinifera*	VvimiPEP164c (vvi-miR164c)	16	N/A	Upregulation	Downregulation of *VvMYBPA1* grapevine transcription factor	Inhibition of proanthocyanidin synthesis and stimulates anthocyanin accumulation	[24]
*Vitis vinifera*	VvimiPEP172b (vvi-miR172b)	16	N/A	Upregulation	Downregulation of *VvRAP2-7-1*	Increase in cold tolerance in grapevine	[25]
*Vitis vinifera*	VvimiPEP3635b (vvi-MIR3635b *)	11	N/A	Upregulation	Downregulation of *VvENT3*	Increase in cold tolerance in grapevine	[25]
*Barbarea vulgaris*	BvmiPEP164b (bv-miR164b *)	8	N/A	Upregulation	Downregulation of *NAC1*, *NAC4*, *NAC5*, *CUC1* and *CUC2*	Inhibitory effect on main root growth and foliar surface	[18]
*Brassica oleacera*	BomiPEP397a (bo-miR397a *)	10	N/A	Upregulation	Downregulation of *LAC2*, *LAC4* and *LAC17*	Stimulation of main root growth and foliar surface	[18]
*Brassica rapa*	BrmiPEP156a (br-miR156a)	33	TAMRA- labeled peptide	Upregulation	N/A	Moderate stimulation of main root growth	[26]
**Animals**							
Human	miPEP200a (hsa-miR-200a)	187	wb; HA fused peptide over-expressed in cells	No regulation	Inhibit the expression of vimentin in cancer cells	Inhibition of the migration of prostate cancer cells	[29,30]
Human	miPEP200b (hsa-miR-200b)	54	wb; HA fused peptide over-expressed in cells	N/A	Inhibit the expression of vimentin in cancer cells	Inhibition of the migration of prostate cancer cells	[29]
Human	miPEP155 (hsa-miR-155)	17	EGFP-fused ORF	No regulation	No regulation	Suppression of autoimmune inflammation by modulating antigen presentation	[30,31]
Human	miPEP497 (hsa-miR-497)	21	N/A	No regulation	No regulation	N/A	[30]
Human	miPEP22 (hsa-miR-22)	57	wb	N/A	N/A	Tumor suppressor	[32]
Human	miPEP133 (hsa-miR-34a)	133	wb	Up-regulation	N/A	Increase in p53 transcriptional activity by disrupting mitochondrial function	[33]
Human	MISTRAV or MOCCI (hsa-miR-147b)	83	Wb; Immuno- fluorescence of over-expressed peptide	No regulation	N/A	Viral stress response, inflammation and immunity	[34,35]
*Drosophila melanogaster*	MSAmiP (dme-miR-iab-8)	9 to 20	EGFP-fused ORF	No regulation	N/A	Involved in sperm competition	[36]
*Drosophila melanogaster*	DmmiPEP8 (dme-miR- 8)	71	wb	No regulation	No regulation	Wing size reduction	[37]
*Mus musculus*	MmmiPEP31 (mmu-miR-31)	44	EGFP-fused ORF and wb	down-regulation	N/A	Suppression of EAE by promoting the differentiation of Treg cells	[38]

*: miR not present in miRbase.

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
