# Peer review of "The Essentials on microRNA-Encoded Peptides from Plants to Animals"

_biomolecules, 2023, doi:10.3390/biom13020206_

Round 1

Reviewer 1 Report

This is an interesting paper describing recently identified peptides encoded by microRNA genes. The review contains information about all peptides discovered in plants and animals by today, and reveals their mechanisms of action. Much attention is paid to the conservation of peptides, as well as to the future direction of research in this field and prospects for the use of these molecules. The review is well structured, scientifically significant, and, unlike other few reviews on miPEPs, also describes miPEPs in animals.

I would suggest adding the data when the information was taken from the miRBase (line 36). I also would like to point out that the mention in the text of Figure 1 is not in place and the transcript of the miPEP abbreviation on line 76 is no longer needed, it was presented earlier on line 53.

Author Response

We thank the reviewer for the suggestions.

As suggested, we add the name of the miR in the table1 when the information is present in miRbase. we indicated by an * when the miR is not listed in miRbase.

In addition we moved Figure 1 from the introduction to the first paragraph.

We also removed the sentence line 76 concerning miPEP abbreviation since it is already present line 53

Reviewer 2 Report

This is a thorough review on a new emerging topic of study.  I think it will be of good general interest to the readers.  The authors cover both animal and plant models in a very fair and relevant way not overstating new discoveries but illustrating interesting areas for further study.  A typo was observed in line 86 where authors should remove the word arose.  

Author Response

We thank the reviewer for his suggestions. We corrected the typo line 86 by removing the word arose.